# Job Insecurity and Company Behavior: Influence of Fear of Job Loss on Individual and Work Environment Factors

**DOI:** 10.3390/ijerph20043586

**Published:** 2023-02-17

**Authors:** Remberto Castro-Castañeda, Esperanza Vargas-Jiménez, Sara Menéndez-Espina, Raúl Medina-Centeno

**Affiliations:** 1Department of Psychology, Coast University Center, University of Guadalajara, Puerto Vallarta 48280, Mexico; 2Department of Education, Padre Ossó Faculty, University of Oviedo, 33008 Oviedo, Spain

**Keywords:** job insecurity, work environment, company behavior, work–life balance, precarious work, mental health

## Abstract

The purpose of this research is to analyze to what extent job insecurity is related to different factors related with quality of work life. Specifically, it refers to the individual (work–family balance, job satisfaction, labor and professional development, motivation at work, and well-being at work) and work environment (conditions and environment and safety and health at work) dimensions of the construct. The sample group consisted of 842 workers (375 men and 467 women), aged between 18 and 68 years, from Bahía de Banderas, Mexico. Pearson correlation coefficients between the different variables were carried out, as well as MANOVA and ANOVA analyses and a linear regression analysis. The results showed that workers with low job insecurity obtained higher scores in work–family balance, job satisfaction, labor and professional development, motivation at work, well-being at work, conditions and environment, and safety and health at work, in relation to workers with moderate and high insecurity. The regression analysis confirmed that individual factors explain 24% and environmental factors 15% of job insecurity. This article makes an approximation to the phenomenon of job insecurity in the Mexican context, where the relationship of this variable with quality of work life is verified.

## 1. Introduction

The Sustainable Development Goals (SDGs) have pointed to economic growth and decent work as a pressing axis in the social and labor reality of the world, especially due to the job insecurity that is becoming natural and official in our capitalist system [1,2,3,4]. These conditions reduce the ability to plan and control all aspects of workers’ lives [5,6], as well as guarantee stability, job promotion and social protection.

From a psychosocial perspective, labor precariousness is a process articulated to the dynamics of the social structures of the new financial capitalism [4,7], which promotes social exclusion and the non-participation of citizens in society [8,9]. Labor precariousness is characterized by being a threat to the continuity and stability of employment [10], generating insecurity and a restriction of the power of workers to defend their job. Precarious work can be measured according to several indicators: insecurity about the continuity of employment, low and insufficient wages for the worker, lack of social protection, loss of labor rights, and restricted liberties, which do not allow social and organizational changes [11]. Likewise, this precarious situation prevents the realization of personal and family future plans due to job instability, which affects families’ financial environments [12,13].

Labor precariousness is linked to precarious work and the subjective perception that the worker has [14,15], producing as a consequence a phenomenon called job insecurity. Greenhalgh y Rosenblatt defined job insecurity for the first time as “Perceived powerlessness to maintain desired continuity in a threatened job situation” [16]. Since that time, there have been a wide variety of studies on this phenomenon, as well as different ways of understanding this fear of job loss. However, all of these approaches are subjective processes, where the worker perceives a threat to his future work, as well as the inability to control the situation [17,18]. It is, therefore, an involuntary and uncontrollable situation [19]. Going into detail about the subjective dimension of this phenomenon, job insecurity has a psychosocial approach in which three levels are related: individual, organizational, and socioeconomic [20]. In the line of these authors, it is not only the anticipation of a loss due to a finite-term labor contract, but there are different contextual factors that influence its development, such as unemployment rates or moments of economic crisis [21,22]. In addition to the causes or risk factors that favor the increase in job insecurity, various investigations have focused on its consequences. These also occur at different levels, which we could divide into personal, organizational, and extra-organizational outcomes.

The experience of job insecurity has repercussions in the personal and organizational sphere. At the individual level, a link has been found with alterations in physical health [23,24,25,26] and mental health [27,28,29], producing depression, anxiety [30], emotional exhaustion [31], and low satisfaction in life [32]. All of these elements have an impact on identity [11] and self-esteem [10,25,33], and they affect the functional life of the family, causing an imbalance between family life and work, reducing work well-being [34,35].

If we focus on the organizational dimension, there is evidence that the worker with job insecurity develops a poor work attitude [11,36,37]. It also reduces their trust in the company and organizational commitment, as well as negatively affecting performance and productivity [1,2,38]. This leads to the development of negative interpersonal behaviors such as workplace bullying, decreased social support, and decreased satisfaction among coworkers. It also causes low performance and effort, thus causing a rise in accidents, non-productive behaviors in their positions, and a decrease in creativity [10]. These elements cause the person to have certain alternative behaviors such as looking for a new job, exploring new training opportunities, and increasing applications for social assistance [36,39]. In turn, at the individual and family level, it increases saving behavior, which has consequences on spending plans and activity in family life [10,25,33,40].

The study of job insecurity has had a greater trajectory in Anglo-Saxon countries, being a phenomenon that still has little research in the Latin American context [10,41,42,43]. Some of the most recent studies use this term as a direct translation of job insecurity, understood as poor working conditions [44]. The same occurs in Spain, where Llosa et al. [9] conceptualized the difference between the two terms (with job insecurity translated into Spanish as “incertidumbre laboral”). However, and having evidenced the influence of the socioeconomic context on the way in which workers develop job insecurity, it is interesting to learn more about this phenomenon in countries such as Mexico.

This study aims to provide an approach to the phenomenon of job uncertainty from the point of view of the quality of work life. Among the multiple approaches to the term, Sirgy et al. [45] define it as “employee satisfaction with a variety of needs through resources, activities, and outcomes stemming from participation in the workplace” (p. 242). Segurado Torres y Agulló-Tomas [46] find four general categories indicating quality of work life: individual indicators (job satisfaction, expectations, commitment, etc.); work environment (working conditions, health and safety, etc.); organization (organizational culture, communication, decision making, etc.); and socio-work environment (living conditions, socio-economic factors, etc.). These authors also show the relationship between job security and quality of working life, which is observed in other research studies [47,48,49]. It can be observed that the quality of work life is a broad concept, which allows the study of different aspects related to the world of work. For this study, the individual and work environment factors will be taken as a reference and the following objectives are proposed: (1) Analyze the relationship of job insecurity with the various individual factors (work–family balance, job satisfaction, labor and professional development, motivation at work, and well-being at work) and environmental factors at work (conditions and environment and safety and health at work); (2) Analyze possible divergences between groups (low, moderate, and high job insecurity) and individual factors and work environment factors; (3) Determine the predictive value of individual factors and environmental factors in job insecurity.

## 2. Materials and Methods

### 2.1. Participants

This study used a non-experimental, cross-sectional and ex post facto design. The sample was probabilistic, being the sampling frame used for the Human Resources payroll list. The people to participate were chosen at random. The participants were workers from the municipality of Bahía de Banderas Nayarit, Mexico. Of a total of 1200 employees who work in this center, a total of 843 people participated, 375 men (44.5%) and 467 women (55.5%), aged between 18 and 68 years, (M = 40.83, DT = 11.42). A sampling error of ± 4%, a confidence level of 99.7%, and a population variance of 0.50 were assumed. 

### 2.2. Procedure

Once the corresponding permits were obtained from the authorities involved, the different departments of the City Council proceeded to apply the instruments. The workers were told that their participation was voluntary and that their responses to the instruments were anonymous.

The study respected the regulations and updates of the Helsinki Declaration, such as ethical values in research on human beings, informed consent, right to information, confidentiality, protection of personal data, non-discrimination, and the possibility of abandoning the study.

### 2.3. Assessment Instruments

A quality of life at work scale (ECVT for its Spanish acronym, “Escala de calidad de vida en el trabajo”) was developed by Patlán Pérez [50]. This instrument consists of 117 items grouped into 4 factors: individual factors, work environment factors, company and work factors, and socio-work environment factors. The scale responses range from 1 (strongly disagree) to 6 (strongly agree). For this investigation, two dimensions were used: the individual and the work environment. In turn, these were divided into different subscales. In the group of individual factors, we found six: work–family balance factor, comprising 7 items (e.g., “after work I can spend time with my family”); the job satisfaction factor, comprising 9 items (e.g., “when I achieve my goals at work, I feel satisfied”); the labor and professional development factor, developed with 8 items (e.g., “in the company I have the opportunity to work and continue studying”); the motivation factor at work, consisting of 7 items (e.g., “in this company I feel motivated by the activities I do”); and the well-being factor at work, consisting of 8 items (e.g., “I feel comfortable in my job”). The group of work environment dimensions consisted of the work environment and conditions factor, integrated by 6 items (e.g., “my workplace is clean, hygienic and healthy”), and the occupational health and safety factor, consisting of 8 items (e.g., “my workplace does not comply with the necessary safety measures”). In the study, the reliability coefficients (Cronbach’s alpha) were 0.85 for work–family balance, 0.90 in job satisfaction, 0.93 in work and professional development, 0.93 in work motivation, 0.85 in well-being at work, and 0.96 in the individual global factor. In the case of the work environment factor, the overall reliability coefficient was 0.89, 0.93 for work conditions and environment, and 0.92 for safety and health at work.

The Job Insecurity Scale (JIS-8) was developed by Pienaar et al. [36] and was adapted to Spanish language by Llosa et al. [37]. This scale measures the presence of job insecurity in workers, and consists of 8 items grouped into two dimensions, affective and cognitive. Its responses range is from 1 to 5 (from totally disagreeing to totally agreeing). The global score of the scale was used for this investigation. Reliability coefficients (Cronbach’s alpha) were 0.84 for affective insecurity, 0.72 for cognitive insecurity, and 0.78 for the global scale.

### 2.4. Data Analysis

First, both univariate and multivariate outliers were detected; the first ones were detected by exploring standardized scores. Following the criteria indicated by Hair et al., [51], outliers were considered to be those whose standardized scores presented an absolute value greater than 6. The second ones were detected by applying the Mahalanobis distance [52]. The coding and analysis of the data was carried out in the statistical package SPSS version 22.

First, clusters of three groups were formed: cases scoring one standard deviation below the mean (range 1 to 2.31) were defined as the low insecurity group (*n* = 124, 14.8%); those scoring one standard deviation above the mean (range 3. 85 to 6) were placed in the high insecurity group (*n* = 113, 13.5%); and cases that were within one standard deviation of the mean (range 2.32 to 3.84) were reclassified into the moderate job insecurity group (*n* = 600, 71.7%). The system reported 5 missing values. Then, an analysis of the Pearson correlations was made to determine the relationship between job insecurity with all the variables under study. Establishing the contrast groups, the MANOVA and ANOVA were calculated to analyze the individual variables and work environment factors. Finally, a linear regression analysis was performed to analyze the predictive value of the different independent variables on job insecurity.

## 3. Results

### 3.1. Correlations

Table 1 shows the correlations between the study variables. Significant correlations were obtained between all of them. Job insecurity correlates negatively with: family–work balance (r = −0.388, *p* < 0.01), job satisfaction (r = −0.464, *p* < 0.01), labor and professional development (r = −0.457, *p* < 0.01), motivation at work (r = −0.433, *p* < 0.01), well-being at work (r = −0.418, *p* < 0.01), conditions and environment at work (r = −0.368, *p* < 0.01), and occupational health and safety (r = −0.234, *p* < 0.01).

### 3.2. Manova and Anova of the Job Insecurity Groups and the Individual Factors and Work Environment Factors

The analysis of variance revealed statistically significant differences between the job insecurity groups with the individual and work environment factors (˄ = 0.978, F(7,780) = 5093.76, *p* < 0.001, n2 = 0.978).

The ANOVA showed significant differences in work–family balance (F(2,796) = 51.61, *p* < 0.001, n2 = 0.115), job satisfaction (F(2,796) = 103.29, *p* < 0.001, n2 = 0.206), labor and professional development (F(2,796) = 82.70, *p* < 0.001, n2 = 0.172), motivation at work (F(2,796) = 110.90, *p* < 0.001, n2 = 0.218), well-being at work (F(2,796) = 110.90, *p* < 0.001, n2 = 0.171), conditions and environment (F(2,796) = 78.53, *p* < 0.001, n2 = 0.165), and safety and health at work (F(2,796) = 18.77, *p* < 0.001, n2 = 0.045) (Table 2).

Bonferroni’s post hoc test (0.05) indicated that workers with low job insecurity obtained statistically higher scores in work–family balance, job satisfaction, work and career development, work motivation, well-being at work, conditions and environment, and safety and health at work, in relation to workers with moderate and high insecurity. It should be noted that workers with high insecurity showed the lowest means in all individual factors and the factors of the work environment with respect to moderate and low insecurity (Table 2).

### 3.3. Predictive Value of Individual Factors and Environmental Factors on Job Insecurity

The results of the linear regression analysis confirmed the predictive value of individual factors and environmental factors on job insecurity (Table 3). On the one hand, individual factors explain 24% and environmental factors 15% of the insecurity, the first one having a higher value than the latter.

Regarding the individual factors, it was found that job satisfaction (*β* = −0.193; *p* < 0.001) and labor and professional development (*β* = −0.114; *p* < 0.001) were statistically significant. In turn, the variables family–work balance, motivation at work, and well-being at work were not significant in the predictive dimension of job insecurity.

Regarding the environmental factors variables, it was verified that the working conditions and environment (*β* = −0.330; *p* < 0.001) and safety and health at work (*β* = −0.175; *p* < 0.001) were statistically significant variables in explaining job insecurity.

## 4. Discussion

In this research work, the purpose of analyzing the relationships between individual factors and environmental factors at work in relation to job insecurity was set. First, the aim was to analyze the relationship of job insecurity with various individual factors (work–family balance, job satisfaction, labor and professional development, motivation at work, and well-being at work) and environmental factors at work (conditions and environment and safety and health at work).

Regarding the first objective, the results confirm that insecurity is significantly related to individual factors and the work environment. The data indicate that workers who experience low job insecurity, compared with moderate and high levels, have a better fit within the individual variables and the work environment. Thus, workers with low job insecurity manage to obtain a balance between work demands and family and personal requirements. It increases feelings of stability, motivation, security, efficiency, productivity, and support. These data are consistent with the investigations by Vargas-Jiménez et al. [29], Menéndez-Espina et al. [15], Lee et al. [25], and Shoss [10]. Likewise, the fact that job insecurity correlates negatively with family functionality [34,35] is relevant. In turn, workers with low insecurity experience job satisfaction, as well as a favorable perception, pleasant and emotionally positive state about the activities they perform in the company [19,32].

Secondly, the objective was to analyze the possible divergences between the groups (low, moderate, and high job insecurity) and the individual and environmental factors at work. In this respect, the results indicate that workers with low job insecurity perceive that the company gives them opportunities to apply and develop their work skills. This allows them to acquire knowledge and develop new useful skills for their performance, as well as promotion to new jobs, contributing to a positive perception of a career in the company. Following this path, low job insecurity is associated with greater motivation at work; that is, a worker who experiences job stability favors the development of desires and expectations to satisfy their personal, work, and professional needs through the performance of their work. This implies promoting feelings of satisfaction, security, motivation, and commitment to their work [1,2,37,38]. As the results also suggest, the group of workers with low insecurity perceives well-being at work, experiencing an affective state of pleasure and activation by their work environment.

Regarding the work environment factor, the group with low job insecurity perceives that they have the technical, social, and labor factors of their work environment, adequate lighting, ventilation, temperature, comfort, cleanliness, and health to achieve an excellent performance, as well as physical and mental well-being. At the same time, employees perceive that the company uses preventive techniques for the protection and elimination of health risks. These include the care of physical integrity and the development of healthy work, promoting safety and health in the work context.

The group of workers with high job insecurity show the worst perception of individual factors and the work environment. Thus, at the individual level, they experience an imbalance between personal and family demands and those of work. This is related to a greater state of insecurity, instability, demotivation and dissatisfaction at work [10,25,33,34,35]. These phenomena are associated with mental health problems such as psychological discomfort caused by a combination of anxiety and depression [11,28,29,30,31,33], as well as physical health alterations [23,24].

In the field of work environment factors, workers who experience high job insecurity perceive that they do not have adequate technical, social, and labor resources in their work environment for their performance and mental and physical well-being. They perceive their place as uncomfortable, unhealthy, and unsafe, with poor lighting, ventilation, and temperature. In short, they consider it an unpleasant place to conduct their work, reducing trust in the company and reducing their performance, efficiency, creativity, and productivity [2,29,37]. In turn, they observe insecurity at work (referring here to insecurity as lack of security, defined as “a set of techniques and procedures whose purpose is to eliminate or reduce the risk of workplace accidents” [53]), with a fear of suffering accidents and having consequences on their health, directly producing poor attitudes of collaboration [11].

Finally, we also set out as the third objective to determine the predictive value of individual and environmental factors on job insecurity. The results show that the individual factors with the greatest explanatory weights are job satisfaction and work and professional development. With respect to work environment, both conditions and environment and health and safety at work predict job insecurity, a systemic cycle of mutual feedback.

### 4.1. Contributions of the Study

This paper can effectively guide those who work in Human Resources departments. One possibility is the design of mental health courses that address both individual factors, as well as job and professional satisfaction and development at work, such as job satisfaction and professional development at work, as well as security conditions and health at work, since they weigh more in the development of job insecurity. At a macrosocial level, this work could guide public services to design policies that impact Federal Labor Law and collective labor contracts with the goal of changing job insecurity and building decent work in the Mexican context. The fact of having carried out this research in Mexico is relevant, as it helps to understand the processes and effects of job insecurity in the Latin American context. Therefore, it represents a contribution to propose labor policies that imply a significant improvement in the incorporation and guarantee of labor rights, proposing more advances in stability, security, and work–life balance.

### 4.2. Limitations and Future Research

It is important to note that the results presented here must be interpreted with caution, due to the cross-sectional and correlational nature of the data, which do not allow establishing causal relationships between the variables. A longitudinal study with measurements at different times would help to clarify the links observed here. It would also be interesting to continue to deepen the knowledge of how the phenomenon of job insecurity occurs in the Latin American social and labor context, making comparisons between countries and exploring other psychological outcomes.

## 5. Conclusions

This paper provides relevant observations on the study of job insecurity and its relationship with various factors related to job conditions and job satisfaction. Alluded to factors are considered as individual, but are involved in the interaction between worker and employment, as well as factors of the work environment. These refer to the conditions and aspects of safety and health. Likewise, it has been possible to observe the differences between people who present higher and lower levels of job insecurity, and also the factors that explain to a greater extent the presence of the fear of job loss.

## Figures and Tables

**Table 1 ijerph-20-03586-t001:** Pearson correlations between the study variables.

	1	2	3	4	5	6	7	8
Job insecurity	1							
2. Work–family balance	−0.388 **	1						
3. Job satisfaction	−0.464 **	0.737 **	1					
4. Labor and professional development	−0.457 **	0.650 **	0.779 **	1				
5. Motivation at work	−0.433 **	0.576 **	0.742 **	0.826 **	1			
6. Well-being at work	−0.418 **	0.640 **	0.758 **	0.708 **	0.726 **	1		
7. Conditions and environment	−0.368 **	0.539 **	0.636 **	0.607 **	0.682 **	0.691 **	1	
8. Safety and health at work	−0.234 **	−0.024	0.030	0.039	0.103 **	−0.003 **	0.179 **	1

** The correlation is significant at the 0.01 level.

**Table 2 ijerph-20-03586-t002:** Differences between groups (low, moderate, high job insecurity) in individual factors and work environment.

	Low Insecurity		Moderate Insecurity		High Insecurity		F
	M	(ST)	M	(ST)	M	(ST)	
**Individual Factors**							
Work–family balance	5.70 ^a^	0.40	5.20 ^b^	0.70	4.75 ^c^	0.87	51.61 ***
Job satisfaction	5.68 ^a^	0.37	5.01 ^b^	0.70	4.29 ^c^	0.98	103.29 ***
Work and professional development	5.42 ^a^	0.74	4.62 ^b^	0.95	3.78 ^c^	1.0	82.70 ***
Motivation at work	5.43 ^a^	0.70	4.62 ^b^	0.95	3.78 ^c^	1.0	110.90 ***
Well-being at work	5.63 ^a^	0.47	4.89 ^b^	0.89	4.08 ^c^	1.1	82.18 ***
**Work Environment Factors**							
Conditions and environment	5.40 ^a^	0.75	4.88 ^b^	0.88	3.90 ^c^	1.2	78.53 ***
Safety and health at work	4.40 ^a^	1.43	4.07 ^b^	1.2	3.43 ^c^	0.98	18.77 ***

Note: M = mean; ST = standard deviations; F = Fisher–Snedecor’s F test; Bonferroni’s F test: a > b > c; *** *p* < 0.001.

**Table 3 ijerph-20-03586-t003:** Results of multiple linear regression analysis of predictors associated with job insecurity (standardized coefficients).

Predictor Variables	*R^2^* Adjusted	F	*β*	*p*
**Individual Factors**	0.24	52.50		
Work–family balance			0.175	−0.063
Job satisfaction			−0.205	0.001 *
Work and professional development			−0.157	0.011 *
Motivation at work			−0.074	0.209
Well-being at work			−0.051	0.317
**Work Environment Factors**	0.15	78.31		
Conditions and environment			−0.330	0.001 *
Safety and health at work			−0.175	0.001 *

Note: R^2^ = squared multiple correlation; F = Fisher–Snedecor’s F test; *β* = Beta; *p* = α = 0.05. * *p* < 0.05.

## Data Availability

Data sharing is not applicable to this article.

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
