# Peer review of "Job Insecurity and Company Behavior: Influence of Fear of Job Loss on Individual and Work Environment Factors"

_ijerph, 2023, doi:10.3390/ijerph20043586_

Round 1
Reviewer 1 Report
“The authors investigated “Job insecurity and company behavior: influence of fear of losing a job on individual labor factors and work environment”. The research problem is very relevant and topical for many reasons. Optimum productivity is critical to full-employment income and hence, national income. Unfortunately, job insecurity is inimical to optimum productivity as it stimulates a reasonable number of employees to spend a reasonable portion of their time thinking of alternative or parallel jobs. Thus, the fear of losing one’s job can precipitate the personalization of corporate time to search for more secure jobs. To this end, the importance of this research problem cannot be overemphasized. Nevertheless, there are some observations that require the authors’ attention for correction to permit the enhancement of the quality of the article and thus, enable it stand the chance of publication in a top quality Journal as this.
(A) Literature Review
(i) Job security and job insecurity as well as labour and environmental factors are well-researched concepts in the literature. Thus, there is abundant literature on the on theses concepts. Unfortunately, the authors failed to provide a robust literature review on the research problem. In fact, literature review is barely available as the authors jumped from the introduction to the materials and method. The failure of the authors to provide a good literature review section is a big dent on the article.
(ii) The authors failed to provide a clear articulation the gap (s) in literature. The fact that not many authors have examined the topic in Latin America is not sufficient. In any case, what is the evidence that the research problem is deficient in empirical literature in Latin America?
(iii) Based on the abundant empirical literature on the research problem, there are theories on job security and job insecurity. Unfortunately, the authors failed to underpin the study with any theory thus denying the readers the opportunity of exposure to the relevant theoretical insights
(B) Methodology
(iv) The Methods section is fundamentally flawed. The authors did not indicate how they determined the sample size of the study. Besides, how were the authors choose the participants? What was the sampling technique? Did the authors earmark all the employees for participation? Was it a case study?
(v) The authors applied a scale that was validated in the Spanish setting to their study in Latin America. Is there any evidence that the usability of the scale will not suffer vitiation by environmental factors in Latin America?
(vi) The authors’ usage of ANOVA and MANOVA is not justified. They said “Establishing the contrast groups, the MANOVA and ANOVA were calculated to analyse the individual variables and work environment factors.” . . These statistics are employed in testing for differences in mean values. How did did the authors use ANOVA and MANOVA to analyse the individual variables and environmental factors?
(vii) The authors used SPSS to code their data. How did they implement the correlation test, ANOVA and MANOVA tests?
(viii) The classification of workers/ job insecurity into low, moderate and high is not explicitly explained. They said “. . . forming three groups: low insecurity, moderate insecurity and high insecurity. Workers who score a standard deviation above the mean are classified as the high insecurity group; those who score one deviation below the mean are placed in the low insecurity group, . . .” How did the workers score standard deviations? Did workers score “standard deviations” or the authors computed the standard deviation of the scores of workers between work categories?
(C) Conclusion
(ix) The conclusion section is poorly done. The authors failed to articulate the section properly
(x) The authors further failed to articulate the contribution of the study properly. This makes the value of the study unclear.
(xi) The work has numerous grammatical shortcomings. The authors must edit the work properly and correct all errors.
Author Response
(A) Literature Review
(i) Job security and job insecurity as well as labour and environmental factors are well-researched concepts in the literature. Thus, there is abundant literature on the on theses concepts. Unfortunately, the authors failed to provide a robust literature review on the research problem. In fact, literature review is barely available as the authors jumped from the introduction to the materials and method. The failure of the authors to provide a good literature review section is a big dent on the article.
We deeply appreciate this comment. We have included new literature about Job Insecurity, as well as the concepts of quality of work life, which include the individual and environmental factors that we have measured in the paper (lines 46-59; 67-85). As a result, the introduction section is much richer.
(ii) The authors failed to provide a clear articulation the gap (s) in literature. The fact that not many authors have examined the topic in Latin America is not sufficient. In any case, what is the evidence that the research problem is deficient in empirical literature in Latin America?
We have included references about the study of job insecurity in Latin America, recent research, and a review about the concept of job insecurity and its translation into Spanish (lines 78-85).
(iii) Based on the abundant empirical literature on the research problem, there are theories on job security and job insecurity. Unfortunately, the authors failed to underpin the study with any theory thus denying the readers the opportunity of exposure to the relevant theoretical insights.
We have expanded the literature review on this topic in the introduction section.
(B) Methodology
(iv) The Methods section is fundamentally flawed. The authors did not indicate how they determined the sample size of the study. Besides, how were the authors choose the participants? What was the sampling technique? Did the authors earmark all the employees for participation? Was it a case study?
We have included this information in lines 108-114.
The sample size is large because it is intended to be representative of the universe with the intention of minimizing error and maximizing confidence and probability. In turn, robust samples are recommended for multivariate analysis. It was calculated according to Arkin & Colton (1985), establishing the number of 726 (99.7%, confidence level and sampling error of ± 4%) and was increased considering the incomplete questionnaires that could possibly be cancelled.
(v) The authors applied a scale that was validated in the Spanish setting to their study in Latin America. Is there any evidence that the usability of the scale will not suffer vitiation by environmental factors in Latin America?
The instrument validated in the Spanish context was the Job Insecurity Scale (JIS-8) by Pienaar et al. which, like all instruments, had its pilot test prior to application. In turn, this instrument has already been used by the authors in other research in the Mexican context (Vargas et al.,2020) and they obtained reliability coefficients equal to those reported, showing adequate psychometric properties, a clear and coherent structure with two dimensions, affective and cognitive insecurity.
The quality of life at work scale (QWL) of Patlán Pérez was developed in Mexico and arose from a review and identification of quality of life at work scales in the literature worldwide. The individual factor and work environment dimensions of this scale were used, so they will not be affected in the dimensions since their origin is in the Latin American context.
(vi) The authors’ usage of ANOVA and MANOVA is not justified. They said “Establishing the contrast groups, the MANOVA and ANOVA were calculated to analyse the individual variables and work environment factors.” . . These statistics are employed in testing for differences in mean values. How did did the authors use ANOVA and MANOVA to analyse the individual variables and environmental factors?
Thank you for your observation. In our view, the use of Anova and Manova is justified in the text. First, workers are classified based on their scores on the job insecutiry scale, with the aim of forming groups or clusters that are homogeneous among themselves and heterogeneous among them.
The text states (line 184) "The analysis of variance revealed statistically significant differences between the job insecurity groups with the individual factors and work environment factors (Ë„ = .978, F (7,780) = 5093.76, p< .001, n2 = .978)"; at the same time, the Anovas with the variables and the Manovas comparing the three groups in Table 2 with the corresponding statistics are pointed out.
(vii) The authors used SPSS to code their data. How did they implement the correlation test, ANOVA and MANOVA tests?
The statistical relationships begin with an exploratory phase, which is the correlation analysis aimed at determining the degree of association or correlation between the variables in the population. Table 1 of the paper correlates job insecurity with individual and work environment variables, using Pearson's correlation coefficient, which measures the degree of bivariate association: when there is a positive correlation between the variables, one variable increases and the other also increases in constant proportion; a negative correlation consists of one variable increasing and the other decreasing in constant proportion. Regarding effect size, Cohen (1988) establishes that when (r= .10) it is considered a low effect size; medium when (r=.30) and large when (r > .50). In the study, significant (at the .01 level) and negative correlations were obtained with job uncertainty with a medium effect. Analysis of variance (ANOVA) is used to compare several groups on a quantitative variable. The hypothesis being tested is that the population means (the means of the dependent variable at each level of the independent variable) are equal. The study indexes that it "revealed statistically significant differences between the job insecurity groups with the individual factors and work environment factors (Ë„ = .978, F (7,780) = 5093.76, p< .001, n2 = .978)"; and it is specified in the different subfactors or variables: "The ANOVA showed significant differences in work-family balance (F, (2,796) = 51.61, p<.001, n2=. 115), job satisfaction (F, (2,796) = 103.29, p<.001, n2=.206), job and career development (F, (2,796) = 82.70, p<.001, n2=.172), job motivation (F, (2,796) = 110.90, p<.001, n2=. 218), well-being at work (F, (2,796) = 110.90, p<.001, n2=.171), conditions and environment (F, (2,796) = 78.53, p<.001, n2=.165) and occupational safety and health (F, (2,796) = 18.77, p<.001, n2=.045) (Table 2).
Multivariate analysis was used to assess the differences between the labor insecurity groups and the multiple dependent variables (individual factors and environmental factors); to test the hypothesis of equality of means, the Fisher-Snedecor’s F statistic was obtained, which reflects the existing resemblance between the means of the groups being compared. Its basis of statistical contrast is the more different the effects of the treatments, the more different the means are expected to be, or inversely. In this study the F statistic is reported in table 2. And the direction of the comparisons of the Bonferroni tests (a>b>c). A note was also added to the table indicating the meaning of the abbreviations used.
(viii) The classification of workers/ job insecurity into low, moderate and high is not explicitly explained. They said “. . . forming three groups: low insecurity, moderate insecurity and high insecurity. Workers who score a standard deviation above the mean are classified as the high insecurity group; those who score one deviation below the mean are placed in the low insecurity group, . . .” How did the workers score standard deviations? Did workers score “standard deviations” or the authors computed the standard deviation of the scores of workers between work categories?
We added an explanation in lines 161-166.
Additionally, in the SSPS program, there is another procedure for the conformation of the Clusters and it is based on the same principle of conforming homogeneous groups from grouping them in a centroid, which are heterogeneous among themselves. The first stage explores the quality of the clusters (two-stage clustering) and then, the second stage is to form the groups by grouping them according to the centroids (K-means clustering).
(C) Conclusion
(ix) The conclusion section is poorly done. The authors failed to articulate the section properly
We have changed the structure of the conclusion, changing the limitations and future research section into the discussion. We have also added some text in conclusions.
(x) The authors further failed to articulate the contribution of the study properly. This makes the value of the study unclear.
We fixed it in the lines 282-293.
(xi) The work has numerous grammatical shortcomings. The authors must edit the work properly and correct all errors.
We have made a deep revision of the language.
References:
Arkin, H., & Colton, R. (1985). Tables for statisticians. Barnes & Noble.
Cohen, J. (1988). Statistical power analysis for the behavioral sciences. Lawrence Erlbaum.
Vargas-Jiménez, E.; Castro-Castañeda, R.; Agulló Tomás, E.; Medina Centeno, R. Job Insecurity, Family Functionality and Mental Health: A Comparative Study between Male and Female Hospitality Workers. Behav. Sci. 2020, 10, 146. https://doi.org/10.3390/bs10100146
Reviewer 2 Report
Please rewrite your abstract part and It does not reflect your research idea. Please revise this part.
Background section, Please find a few articles related to your area of interest. You need to highlight your problem based on the fact. When I read your background of the study, the problem is missing. What are the related issues related to the insecurity of jobs?
In the LR.
Please explain in detail, Start with the research gap after need to discuss your DV variable, then explain the IV. Find the most important article related to job security and link it with your finding.
I will strongly suggest to the researcher please further improve the empirical analysis part; descriptive information is insufficient to justify the publication.
The researchers have not discussed this study's theoretical and practical contributions. The author (s) should discuss this study's theoretical and practical contribution in the separate subsection under discussion for more clarity.
The author should provide a precise conclusion section.
Author Response
Please rewrite your abstract part and It does not reflect your research idea. Please revise this part.
We greatly appreciate the suggestion of improving the abstract, we rewrote it.
Background section,
Please find a few articles related to your area of interest. You need to highlight your problem based on the fact. When I read your background of the study, the problem is missing. What are the related issues related to the insecurity of jobs?
We deeply appreciate this comment. We have included new literature about Job Insecurity. We have also included references about the study of job insecurity in Latin America, recent research, and a review about the concept of job insecurity and its translation into Spanish. As a result, the introduction section is much richer [lines 46-59; 67-85]
In the LR.
Please explain in detail, Start with the research gap after need to discuss your DV variable, then explain the IV. Find the most important article related to job security and link it with your finding.
We appreciate this suggestion. In addition to a major review of the literature on job insecurity, we have also included additional information about the quality of work life, which include the individual and environmental factors that we have measured in the paper (lines 86-98)
I will strongly suggest to the researcher please further improve the empirical analysis part; descriptive information is insufficient to justify the publication.
We have included information about the sample, and explained depper the clusters (lines 155-171).
The researchers have not discussed this study's theoretical and practical contributions. The author (s) should discuss this study's theoretical and practical contribution in the separate subsection under discussion for more clarity.
We have discussed the contributions of the research. However, we have marked it more explicitl (lines 281-293).
The author should provide a precise conclusion section.
We have changed the structure of the conclusion, changing the limitations and future research section into the discussion. We have also added some text in conclusions.
Reviewer 3 Report
I rate the job as good. This is a scientific discussion of job insecurity.
The work structure is appropriate. I really appreciate the research.
The authors did a good discussion, outlined the limitations of the study, and compared it with other studies.
I propose to improve the introduction, which is the weakest part of the article. Thus, additional literature items should be added.
Congratulations on your work
Author Response
I rate the job as good. This is a scientific discussion of job insecurity.
The work structure is appropriate. I really appreciate the research.
The authors did a good discussion, outlined the limitations of the study, and compared it with other studies.
I propose to improve the introduction, which is the weakest part of the article. Thus, additional literature items should be added.
We greatly appreciate your review, and we are pleased that you find it correct.
We have included new literature about Job Insecurity. We have also included references about the study of job insecurity in Latin America, recent research, and a review about the concept of job insecurity and its translation into Spanish. As a result, the introduction section is much richer [lines 46-59; 67-85]. We also added information about the concept of quality of work life, which include the individual labor factors and work environmental factors (86-98).